# Bacterial genome size and gene functional diversity negatively correlate with taxonomic diversity along a pH gradient

Cong Wang [1,2], Qing-Yi Yu[1,2], Niu-Niu Ji[1,3], Yong Zheng [1,4], John W. Taylor [5], Liang-Dong Guo [1,2] ✉ & Cheng Gao [1,2] ✉

Bacterial gene repertoires reflect adaptive strategies, contribute to ecosystem functioning and are limited by genome size. However, gene functional diversity does not necessarily correlate with taxonomic diversity because average genome size may vary by community. Here, we analyse gene functional diversity (by shotgun metagenomics) and taxonomic diversity (by 16S rRNA gene amplicon sequencing) to investigate soil bacterial communities along a natural pH gradient in 12 tropical, subtropical, and temperate forests. We find that bacterial average genome size and gene functional diversity decrease, whereas taxonomic diversity increases, as soil pH rises from acid to neutral; as a result, bacterial taxonomic and functional diversity are negatively correlated. The gene repertoire of acid-adapted oligotrophs is enriched in functions of signal transduction, cell motility, secretion system, and degradation of complex compounds, while that of neutral pH-adapted copiotrophs is enriched in functions of energy metabolism and membrane transport. Our results indicate that a mismatch between taxonomic and functional diversity can arise when environmental factors (such as pH) select for adaptive strategies that affect genome size distributions.

For decades, the use of ribosomal RNA (16S rRNA) sequence to identify bacteria, infer their phylogenies, and characterize their communities has revolutionized our understanding of the biodiversity, biogeography, and ecology of microbiomes. One of the most important discoveries made using the 16S rRNA gene as a biomarker has been the observation that bacterial diversity peaks at neutral pH as compared to acid pH[1], and in many subsequent studies it has been proven to be one of the most consistent features of microbial communities (Supplementary Table 1). However, this pH-diversity relationship cannot be directly embedded into classic ecological models such as the biodiversity-ecosystem functioning framework, unless the relationship between taxonomic and functional diversities is clarified.

Functional traits can reflect response and adaptation of microbes to changes in resources and stress[2]. Soil pH has been recognized as the essential driver of microbial structure and function[3–5], and changes of soil pH at regional and latitudinal scales are coupled with changes in both resources and stress[6,7]. For example, acid soil of natural tropical forests can be a harsh environment with low resource availability, whereas neutral pH soil of natural temperate forests can be a benign environment with high resource availability[8,9]. Bacteria adapted to these different soils may share some functional traits, but show variation in other traits, such as genome size, 16S rRNA gene copy number, GC content, and growth rate[10]. Variation in genome size can involve gene diversity and functional versatility[11–13], because the number of genes of a bacterial genome is strongly positively correlated

[1]State Key Laboratory of Mycology, Institute of Microbiology, Chinese Academy of Sciences, 100101 Beijing, China. [2]College of Life Sciences, University of Chinese Academy of Sciences, 100049 Beijing, China. [3]Center for Advanced Bioenergy and Bioproducts Innovation, University of Illinois Urbana-Champaign, Urbana, IL 61801, USA. [4]School of Geographical Sciences, Fujian Normal University, 350007 Fuzhou, China. [5]Department of Plant and Microbial Biology, University of California, Berkeley, CA 94720, USA. ✉e-mail: guold@im.ac.cn; gaoc@im.ac.cn

with genome size[14,15]. Previous studies suggest that slow-growing oligotrophs living in resource-scarce soil may carry large genomes[16,17], as do stress-tolerators found in harsh environments when compared to non-tolerators in benign environments[11,13]. Slow-growing oligotrophs in resource-poor habitats are often characterized by fewer copies of 16S rRNA genes than fast-growing copiotrophs found in high-carbon, resource-rich habitats[10]. GC content also varies with environment, with bacteria having lower GC content being more abundant in the nutrient-deficient surface than in the nutrient-rich interior of the ocean[18,19]. Guided by these studies, we hypothesize $H_1$ that the oligotrophic, stress-tolerators found in acid environments will have larger genomes than the copiotrophic, non-tolerators found in neutral pH environments.

Functional gene diversity of communities does not necessarily reflect taxonomic diversity because community-level, functional gene diversity is influenced by both taxonomic diversity and average genome size. An example where both taxonomic and functional diversities of bacteria are positively correlated is provided by Bahram et al.[7], who researched soils along an environmental gradient of increasing pH and latitude. Conversely, Kerfahi et al.[20] found that the diversity of functional genes was negatively correlated with pH of soils collected along a gradient of increasing altitude. Explaining the negative correlation of functional gene diversity and increasing environmental pH, given the consensus that bacterial taxonomic diversity increases from acid to neutral pH[1] (Supplementary Table 1), may require consideration of genome size. As suggested by our $H_1$, small genomes are likely to be dominated by indispensable, core genes[21,22], whereas large genomes can harbor the core genes as well as genes of diverse function[10]. As a result, a relatively low diversity community of large genome taxa can support a higher diversity of functional genes than a high diversity community of small genome taxa. Therefore, we hypothesize $H_2$ that along a gradient of increasing pH, there will be a mismatch between increasing bacterial taxonomic diversity and decreasing functional diversity.

The traits responsible for the adaptations that underlie the spectrum of adaptive strategies from oligotrophic communities to copiotrophic communities include growth rate, carbon use efficiency, biomass yield, enzyme production, genome size and 16S rRNA gene copy number[23–26]. Recently, our understanding of bacterial adaptive strategy has been improved by the inclusion of genome information, however, these improvements challenge some assumptions about associations of traits with communities of oligotrophic or copiotrophic bacteria[10,25,27,28]. Recognizing that our understanding of the gene repertoire underpinning oligotrophy and copiotrophy is in its infancy, we hypothesize, $H_3$, that the gene repertoire of oligotrophic and copiotrophic communities along an environmental pH gradient will differ in traits of cell motility, chemotaxis, secretion systems, resource transporters, and defense.

To test the three hypotheses proposed here, we analyzed the diversity of bacterial taxa using 16S rRNA gene amplicons and the diversity of functional genes using shotgun metagenomes sequenced from DNA extracted from 36 plots of 12 forests along a latitudinal gradient (N21.6°–N50.9°) in China, where soil pH ranged from 3.68 to 7.22 (Fig. 1a). Our analysis supported $H_1$ because genome size decreased as soil pH rose from acid to neutral. Second, $H_2$ was supported as taxonomic and functional gene diversities were significantly negatively correlated. Finally, $H_3$ was supported as the gene repertoires of acid-adapted, oligotrophic strategists were enriched in signal transduction, cell motility, secretion system, and degradation of complex compounds, while those of neutral pH-adapted, copiotrophic strategists were enriched in functions of energy metabolism and membrane transport.

## Results

As a prelude to hypothesis testing, we explored the role of soil pH in shaping the compositions of bacterial communities and functional genes. First, principal coordinate (Pco) analysis showed that both bacterial taxonomic and functional gene compositions were divergent between forests with acid pH soils and forests with neutral pH soils, and envfit analysis showed that soil pH correlated strongly with variation in the composition of bacterial communities ($R^2 = 0.747$, $P < 0.001$) and functional genes ($R^2 = 0.868$, $P < 0.001$) (Fig. 1b, c, Supplementary Fig. 1, and Supplementary Data 1). Furthermore, a loss of *Acidobacteria* abundance as soil pH increased from acidic to neutral was seen from sequence of both 16S rRNA gene amplicons ($R = −0.674$, P = 3.925e-06) and metagenomes ($R = −0.777$, $P = 1.52e-08$) (Fig. 1d, e).

### Testing $H_1$: bacterial genome size will decrease from acid to neutral pH

To test our $H_1$, we estimated, from metagenomic data, community-level average genome size using the MicrobeCensus pipeline[29] and estimated average 16S rRNA gene copy number using the method of Pereira-Flores et al.[15]. These estimates showed that bacterial average genome size was strongly negatively correlated with soil pH ($R^2 = 0.423$, $P < 0.001$; Fig. 2a), and that bacterial average 16S rRNA gene copy number was not significantly correlated with soil pH ($R^2 = 0.063$, $P = 0.076$; Fig. 2b). We then estimated bacterial GC content (GC%) using Quast software[30] and found that it increased with soil pH ($R^2 = 0.146$, $P = 0.012$; Fig. 2c); while bacterial growth rate as estimated by gRodon[27] was unrelated to soil pH ($R^2 = 0.014$, $P = 0.489$; Fig. 2d). Besides, both bacterial average genome size and GC% are correlated with several biotic and abiotic variables leading by available Ca (Supplementary Fig. 8).

The detected changes in bacterial genome size with soil pH may be attributed to shifts in stress and resource, i.e., from an acidic, resource-poor, harsh environment to a neutral, resource-rich benign environment. The larger genome size seen in resource-poor, acidic environments has been proposed to accommodate a diversity of genes needed to cope with abiotic stress and resource scarcity and diversity[11,13,16,17]. Small genome size and a high 16S rRNA gene copy number can be linked to fast growth and high resource turnover in resource-rich, benign environments[10,12]. The increase of GC% with increasing pH is consistent with a copiotrophic lifestyle in neutral soils, possibly because GC base pairs require more nitrogen than AT base pairs[18,19]. In summary, our analysis supports $H_1$ because the bacterial genome size decreases as the environment becomes less acidic.

In order to challenge our finding that bacterial genome size, calculated from the MicrobeCensus pipeline[29], is larger in acidic than in neutral pH environments (Fig. 2a), we additionally calculated the community-weighted genome size by referencing the 16S rRNA gene amplicon dataset against the Genome Taxonomy Database (GTDB)[31]. The results again showed that the average genome size of the bacterial community was significantly, negatively correlated with soil pH (Supplementary Fig. 2). The decrease of bacterial genome size was largely caused by changes with increasing soil pH of the most abundant bacterial taxa (Supplementary Fig. 3), including the gain of a genus with a small genome, DA101 (2.80 Mb), and the loss of large-genome taxa of Ca. *Solibacter* (5.52 Mb), Ca. *Koribacter* (5.65 Mb), *Burkholderia* (8.57 Mb), and *Salinispora* (5.56 Mb)[31] (Supplementary Fig. 3). One of the declining taxa, *Xiphinematobacter*, harbors a very small genome (0.91 Mb, likely related to its parasitism of nematodes[32]), but was far less abundant than DA101 and too rare to affect the trend (Supplementary Fig. 3).

The generality of our finding that bacterial genome size decreased from acid to neutral pH in China was tested by

re-analyzing the previously published global dataset of Bahram et al.[7]. Using both the MicrobeCensus annotation of metagenomes and the matching of 16S rRNA amplicons with GTDB, our reanalysis of Bahram et al.[7] again found that bacterial average genome size was negatively correlated with soil pH (Supplementary Fig. 4).

Interestingly, for the datasets of both this study and that of Bahram et al.[7], average genome size significantly, negatively correlated with GC% based on analysis of the metagenome (Supplementary Figs. 5 and 6); however, the average genome size positively correlated with GC% when based on the matching of 16S rRNA community with GTDB (Supplementary Figs. 5 and 6). This obvious, methodological bias should stimulate research on the causes and consequences of the different results in microbial traits detected from the genomic-based and metagenomic-based methods.

## Testing H₂: increasing bacterial taxonomic diversity correlates negatively with decreasing functional diversity along a pH gradient

We tested H₂ by analyzing our 16S rRNA gene amplicon data using the USEARCH pipeline[33] and annotating our metagenome data with the KO database[34]. We then calculated richness and the Shannon diversity index for bacterial taxonomic and functional diversities in each sample and plotted bacterial taxonomic and KO diversity against soil pH. The results showed that bacterial taxonomic diversity increased from acid to neutral pH (Fig. 3a and Supplementary Figs. 7–10), as demonstrated frequently in previous studies (Supplementary Table 1). However, we found that bacterial KO diversity decreased from acid to neutral pH (Fig. 3b). The analysis of the results, showing that taxonomic diversity increased as functional diversity decreased with increasing soil pH, supported a

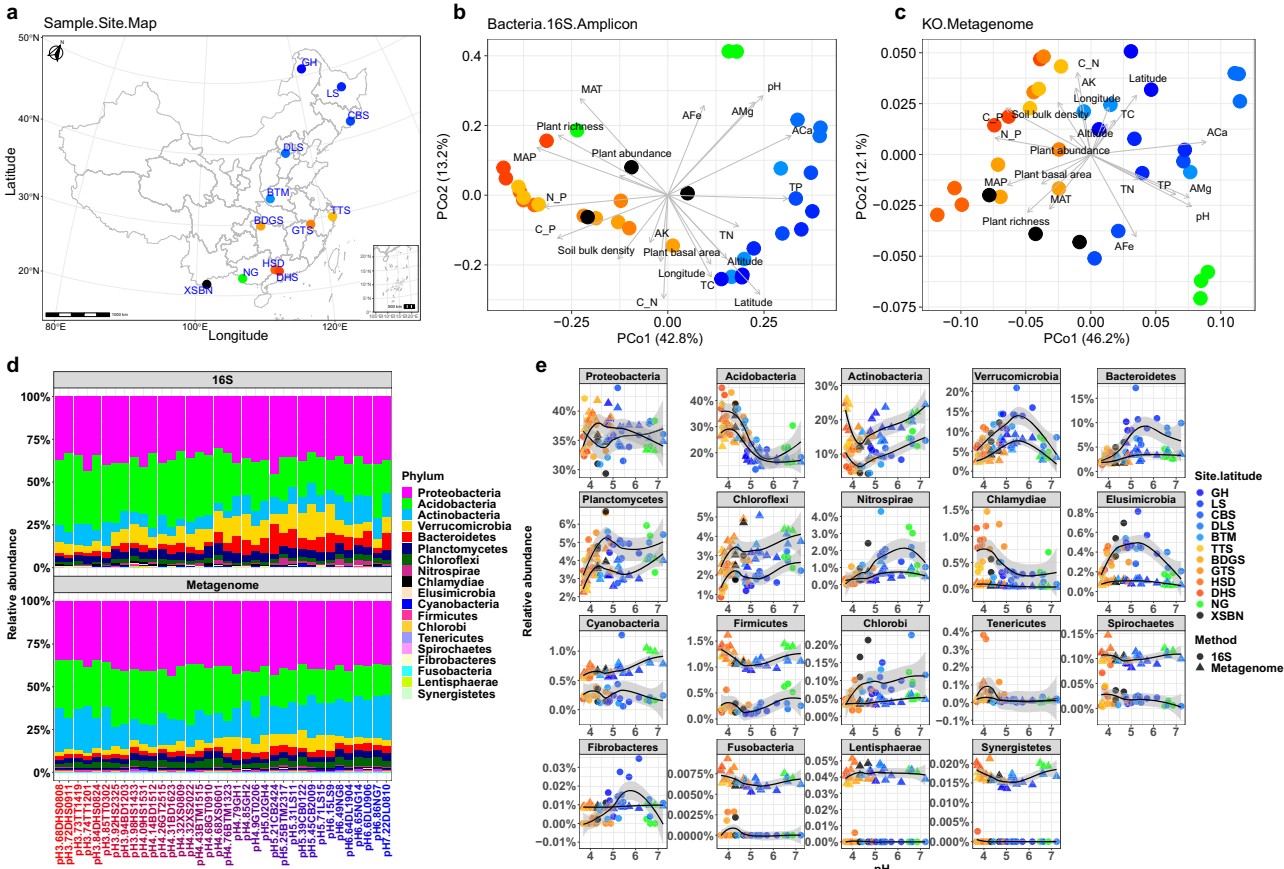

**Fig. 1 | Bacterial community structure along a pH gradient. a Distribution of 12 forests.** Location of 12 forests in China along a latitudinal gradient. GH Genhe, LS Liangshui, CBS Changbaishan, DLS Donglingshan, BTM Baotianman, TTS Tiantongshan, BDGS Badagongshan, GTS Gutianshan, HSD Heishiding, DHS Dinghushan, NG Nonggang, XSBN Xishuangbanna. Color codes for sites in (**a–c, e**) is consistent with other figures of this paper. The data for map was download from DATAV.GeoAltas (http://datav.aliyun.com/portal/school/atlas/area_selector) and visualized by ggplot2 package (https://ggplot2.tidyverse.org/). **b, c** Bacterial community composition in association with environmental variables. Principal coordinate (PCo) analysis with environmental fitting (envfit) showing association of 16S rRNA gene amplicon-based bacterial taxonomic composition (**b**) and metagenome-based bacterial functional composition (**c**) with soil pH as well as other biotic and abiotic variables (the arrowed lines). The strength and significance of association between PCo vectors and variables are provided in Supplementary Fig. 1. MAP mean annual precipitation, MAT mean annual temperature, TC total carbon, TN total nitrogen, TP total phosphorus, ACa

available calcium, AMg available magnesium, AFe available iron, AK available potassium, C_N carbon nitrogen ratio, C_P carbon phosphorus ratio, N_P nitrogen phosphorus ratio. **d** Bacterial phylum composition along a pH gradient. **e** Regression curve of detected bacterial phyla against soil pH. Linear regression model with two-sided test was used for the statistical analysis, and adjusted R-squared was used. Both the 16S rRNA gene amplicon ($R = -0.674$, $P = 3.925e-06$) and metagenome ($R = -0.777$, $P = 1.52e-08$) showed the loss of *Acidobacteria* with increasing soil pH. The relative abundance of *Actinobacteria*, *Planctomycetes*, *Chloroflexi* tends to increase with increasing pH. The relative abundances of *Verrucomicrobia*, *Bacteroidetes*, *Nitrospirae*, *Elusimicrobia* and *Fibrobacteres* peak at around pH 5.5. Note that estimates of 16S rRNA gene amplicon and metagenome sequencing are not necessarily consistent. The relative abundance of *Proteobacteria* remains almost unchanged across the pH gradient. $n = 36$ samples. The grey area around the smooth line indicates the 95% confidence interval. Source data are provided as a Source Data file.

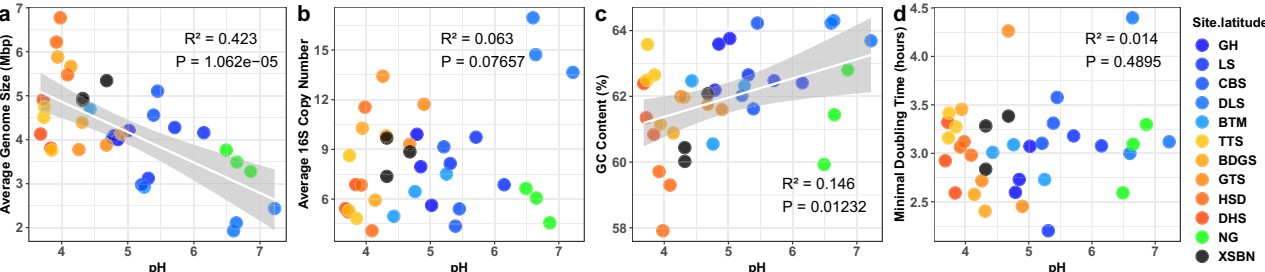

**Fig. 2 | Metagenomic traits along a pH gradient. a** Bacterial average genome size (AGS) decreased as pH changed from acid to neutral. Bacterial AGS are detected by analyzing shotgun metagenome using MicrobeCensus pipeline. **b** Bacterial average 16S rRNA gene copy number (ACN) not significantly associated with soil pH. Bacterial ACN are detected by analyzing shotgun metagenome using the method from ref. 15. **c** Bacterial GC content (GC%) increased as pH changed from acid to neutral. Bacterial GC% are detected by analyzing shotgun metagenome using Quast software. **d** Bacterial estimated growth rate was unaffected by soil pH. Bacterial growth rate (minimal doubling time) was detected by analyzing shotgun metagenome using the gRodon pipeline. Linear regression model with two-sided test was used for the statistical analysis, and adjusted R-squared was used. $n = 36$ samples. The grey area around the smooth line indicates the 95% confidence interval. Source data are provided as a Source Data file.

significantly negative correlation between the two forms of diversities (Fig. 3e). Therefore, H₂ is supported because a positive association between taxonomic and functional diversities is not detected along the pH gradient.

The mismatch between taxonomic and functional diversities may be attributed to the changes in genome size along the pH gradient. Finding that genome size decreased from acid to neutral pH, coupled with previous findings that suggest a positive correlation between bacterial genome size and gene number[35], leads to the expectation of more genes in communities in acid soil compared to those in neutral pH soil. Given that all taxa should share the core of indispensable genes[21,22], a small community of large genome taxa could harbor more genes than a large community of small genome taxa. Indeed, our results showed that bacterial average genome size was positively correlated with KO richness (Fig. 3f), but negatively correlated with bacterial taxonomic diversity (Supplementary Fig. 11).

Our result, including all types of genes, is in line with a recent report by Kerfahi et al.[20], who found that microbial functional gene diversity was negatively correlated with soil pH along a gradient of increasing altitude. Our result is also in line with a previous study focusing on a specific gene group, antibiotic resistance (AR) genes. With AR genes, Delgado-Baquerizo et al.[36] found a negative correlation between soil pH and AR gene diversity for 1012 soil samples collected globally. To compare our approach to these results, we annotated AR genes from our metagenome data using the Resfams database[37], and used these data to show that soil pH was significantly negatively correlated with diversity of AR genes (Fig. 3c and Supplementary Fig. 7). We then extended our analyses of specific gene groups to genes encoding carbohydrate-active enzymes (CAZy genes)[38], again found that soil pH significantly negatively correlated with the diversity of CAZy genes (Fig. 3d and Supplementary Fig. 7). Moreover, diversities of both AR and CAZy genes were also positively correlated with average genome size (Fig. 3g, h and Supplementary Fig. 7).

**Testing H₃: gene repertoire reflects functional trait differences between oligotrophic bacteria adapted to acid pH environments and copiotrophic bacteria adapted to neutral pH environments**

To investigate the distribution of bacterial specific functional traits across the gradient from acidic to neutral pH, we used the co-occurrence network to investigate relationships among KOs. Firstly, we annotated our metagenome data using the KO database to find 11,065 KOs (Supplementary Fig. 12). We then filtered for KOs that occurred in >18 of all 36 samples, yielding 7717 KOs. Co-occurrence among annotated KOs was detected using pairwise Spearman's correlations. These correlations were filtered by Spearman's rho >0.6 and FDR $P < 0.05$ and used to construct a co-occurrence network

composed of 7481 vertices and 1,359,406 edges (Fig. 4a). The most startling features of the co-occurrence network are two, large modules: module 1 (M1) containing 2777 vertices, and module 2 (M2) containing 4309 vertices (Fig. 4a). Having detected these two modules, we then investigated the relationships between soil pH and KOs in them, finding that soil pH negatively correlated with the larger module (M2, 4309 vertices, conditional $R^2 = 0.953$, $F_{1,150814} = 1825$, $P < 0.001$) and positively correlated with the smaller module (M1, 2777 vertices, conditional $R^2 = 0.930$, $F_{1,97194} = 8169$, $P < 0.001$) (Fig. 4b, c).

We then searched within the modules for enriched functions, finding that the acid module (M2) was enriched many key functions: bacterial secretion system, cell motility, xenobiotics biodegradation and metabolism, signal transduction (two component system), metabolism of terpenoids and polyketides, glycan biosynthesis and metabolism, porphyrin metabolism, synthesis of siderophore, and synthesis of lipopolysaccharide (Fig. 4d, e). Correlating specific genes annotated to these functions with environmental variables showed, as expected, that a remarkably large number of genes were negatively correlated with soil pH (Supplementary Figs. 13–15). These genes are involved with essential functions such as motility, bacterial secretion system, and xenobiotics biodegradation and metabolism, a finding largely in line with that of Ramoneda et al.[3] (Supplementary Table 2 and Supplementary Data 2). These results are also consistent with the recent finding that extracellular enzyme investment in substrate acquisition is higher in acid than in neutral soils[39].

Turning to the neutral model (M1), we found enriched functions of energy metabolism, membrane transport, citrate cycle, glyoxylate and dicarboxylate metabolism, and metabolism of amino acids (Fig. 4d, e). Analysis of specific genes for these functions again revealed, as expected, a prevalence of positive correlations with soil pH (Figs. 4 and 5 and Supplementary Figs. 16–18). Our detection of positive associations of pH with essential functions such as carbon metabolism and membrane transport is, again, largely consistent with the findings of Ramoneda et al.[3] and Malik et al.[24] (Supplementary Table 2 and Supplementary Data 2). These results are also in line with the recent findings of higher microbial turnover rate and carbon use efficiency in neutral soil[24,26,39].

The detection of differentially enriched functions in the acid (M2) and neutral (M1) modules enables us to speculate about the adaptive strategy of microbiomes along pH gradients. Our speculations follow a perspective by Malik et al.[25], who suggested that microbes in less stressful, resource-abundant, neutral pH environments would be characterized by functions of carbon metabolism, while those in resource-limited (e.g., acid) environments would have more microbial transporters. As predicted by these authors, we found that microbes adapted to neutral pH environments were enriched for energy

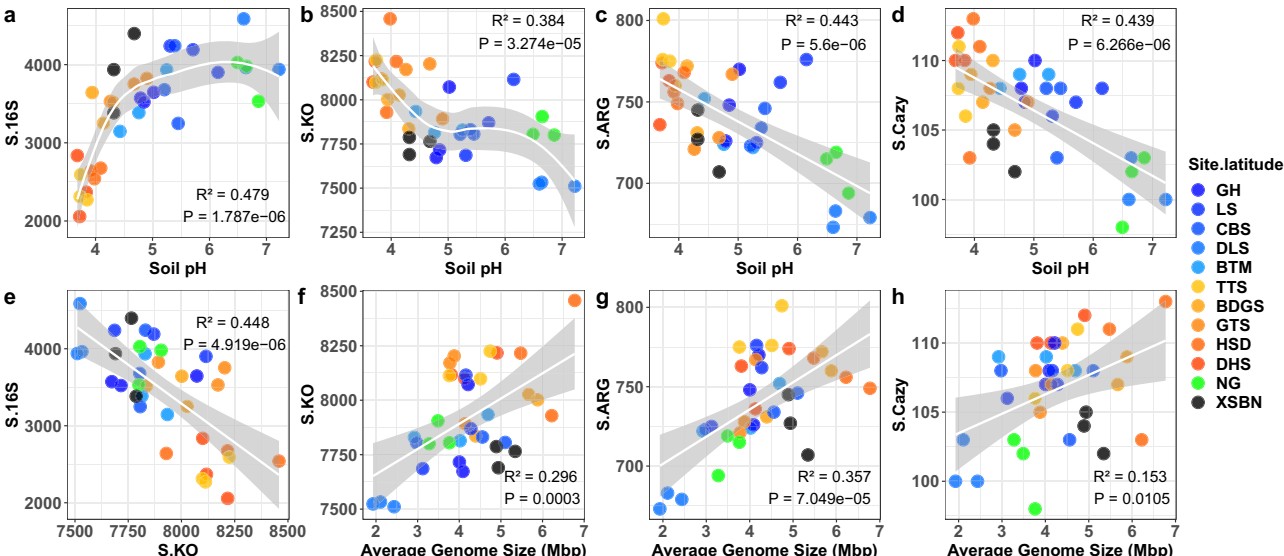

**Fig. 3 | Contrasting distribution patterns for diversities of bacterial taxonomy and functional genes along a pH gradient.** Diversities shown are measured by richness (S), and diversities measured by Shannon's index (H′) are provided in Supplementary Fig. 7. **a** Bacterial taxonomic diversity (S.16S) increased as soil pH changed from acid to neutral. Bacterial operational taxonomic units (OTUs) are detected by 16S rRNA gene amplicon metabarcoding sequencing. **b** Bacterial functional diversity (S.KO) decreased as pH changed from acid to neutral. Bacterial functions are determined from the shotgun metagenome, as annotated by Kyoto Encyclopedia of Genes and Genomes (KEGG) Ontology (KO). **c** Bacterial diversity of antibiotic resistance genes (S.ARG) decreased as pH changed from acid to neutral. Bacterial antibiotic resistance genes are detected based on shotgun metagenome annotated by the Resfam database. **d** Bacterial diversity of carbohydrate-active enzymes (S.Cazy) genes decreased as pH changed from acid to neutral. Bacterial carbohydrate-active enzymes genes are detected based on shotgun metagenome annotated by database of CAZy. **e** Bacterial taxonomic richness (S.16S) negatively correlated with functional gene diversity (S.KO). **f–h** Bacterial average genome size (AGS) positively correlated with functional diversities as measured by **f** S.KO, **g** S.ARG and **h** S.Cazy. Linear regression model with two-sided test was used for the statistical analysis, and adjusted R-squared was used. $n = 36$ samples. The grey area around the smooth line indicates the 95% confidence interval. Source data are provided as a Source Data file.

generation. In neutral pH environments, we also found enrichments for functions related to fast growth, such as resource importation and energy metabolism, indicative of copiotrophic communities. Several previous studies based on rRNA gene copy number estimation also supported a copiotrophic strategy for microbiomes in resource-abundant environments[40,41]. However, not all of our findings are in agreement with previous reports, for example, Malik et al.[25] suggested that there would be more microbial transporters in resource-limited environments, whereas we found more transporters in the resource-abundant environment.

Turning to module 2, our results indicate that microbes adapted to more stressful, resource-limited, acid pH, environments are oligotrophic strategists, based on the enrichments of functions related to resource scavenging and stress tolerance, such as, signal transduction, cell motility, secretion system, and degradation of complex compounds. Two of these traits, motility and degradation of complex substrates, were suggested for microbes in resource-limited environments by Malik et al.[25], as well as in the publication on trait dimensions of Westoby et al.[10], which highlighted the importance of signal transduction in functional versatility. Considering the functions enriched at acid pH, signal transduction is essential to perceive the gradients of resources and stress in the environment. Secretion enables microbes to produce and excrete extracellular enzymes that support degradation of complex compounds in resource-limited environment. Lastly, motility may benefit microbes in terms of resource acquisition and stress defense/avoidance[42–44]. To summarize, our H3 is supported because pH-driven copiotroph- and oligotroph-strategists differ in gene repertoires involving energy metabolism, membrane transport, chemotaxis, motility, secretion system, signal transduction, and defense (Fig. 4).

## Discussion

Our study shows that decoupling between taxonomic and functional diversity can happen when environmental factors (such as pH) select

for life history strategies that influence genome size distributions. The detected changes to genome size at the community-level derive from taxonomic changes along the pH gradient, i.e., a gain of small genome taxa and a loss of large genome taxa from acid to neutral pH. This taxonomic change showed responses to specific functional adaptations along a pH gradient, where bacterial taxa in acid pH soils are enriched in functions of signal transduction, cell motility, secretion system, and degradation of complex compounds, but bacterial taxa in neutral pH soils are enriched in functions of energy metabolism and membrane transport.

Our results challenge the longstanding paradigm that bacterial diversity peaks at neutral pH[1] by going beyond taxonomic diversity to also consider bacterial functional gene diversity which is closely linked to genome size. Finding that taxonomic and functional diversities exhibited contrasting patterns along the pH gradient, our result raises serious questions about the relative contribution of these different types of bacterial diversity to ecosystem function. We found that variation in genome size influenced the relationship between taxonomic and functional diversity in systems where resource availability was poor, as in acid pH soil, or rich, as in neutral pH soil. However, it is known that resource availability can be low in neutral pH soil and high in acid pH soil[6], showing that studies combining taxonomic diversity with functional diversity should be extended to additional ecosystems and additional microbial communities. Thus, our understanding of the structure and function of soil microbiomes would be advanced by a thorough investigation of the functional traits, diversity and adaptive strategy of microbiomes that represent the four quadrants defined by the axes of resource and stress (Fig. 6). A group of microbes in need of such studies is fungi. It is reported that the genome size of fungi also varies by ecosystem[45], but fungal gene numbers may be less predictable due to the existence of non-coding regions.

Expanding the characterization of microbial communities by adding functional gene diversity to taxonomic diversity opens the

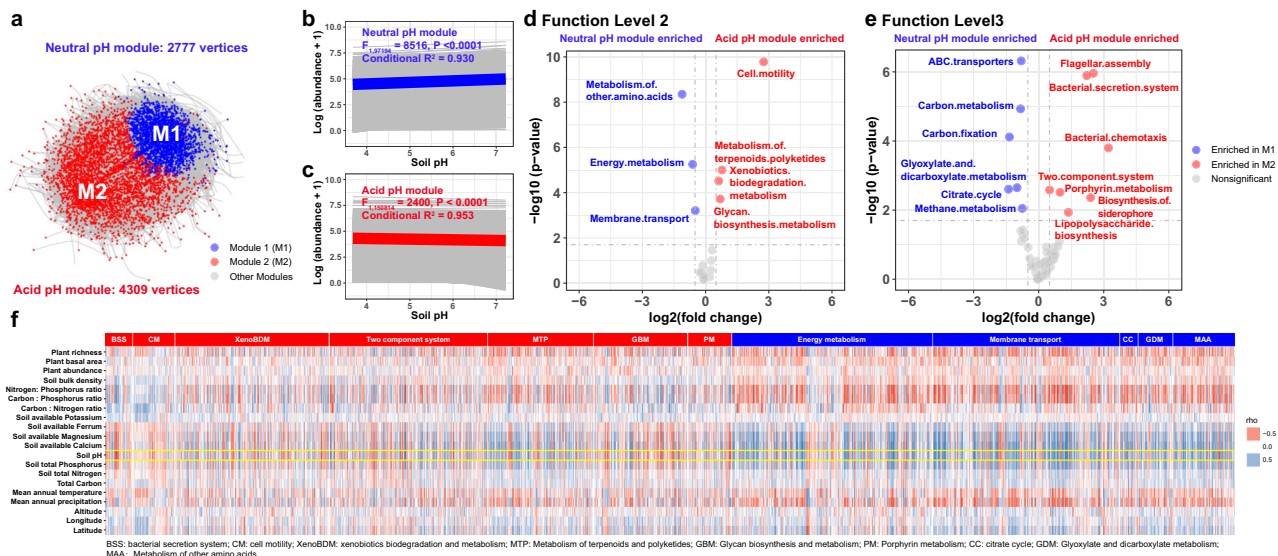

**Fig. 4 | Functional enrichment and depletion along the pH gradient. a** Network analysis of KOs detected two dominant modules, Neutral pH Module, M1 (blue), containing 2777 vertices, and Acid pH module, M2 (red), containing 4309 vertices. **b**, **c** M1 correlates positively and M2 correlates negatively with soil pH. Linear mixed-effects models with two-sided test were used for the statistical analysis. The grey lines are regression lines for each KO, and the colored line is regression line for the average of all KOs. **b** Neutral pH module (M1, blue), showing that soil pH is significantly, positively correlated with M1 and **c** Acid pH module (M2, red), showing that soil pH is significantly, negatively correlated with M2.

**d**, **e** Enrichments of genes in KEGG pathways for Neutral pH (M1, Blue) and Acid pH (M2, Red) modules. Differential expression analysis with two-sided test were used for the statistical analysis. M1 functions were enriched for energy metabolism,

membrane transport, citrate cycle, and glyoxylate, dicarboxylate and amino acid metabolism. M2 functions were enriched for bacterial secretion system, cell motility, xenobiotics biodegradation and metabolism, two component systems, metabolism of terpenoids and polyketides, glycan biosynthesis and metabolism, starch and sucrose metabolism, porphyrin metabolism, siderophore and lipopolysaccharide synthesis. **f** Association between environmental variables and specific genes involved in essential functions drawn from functional pathways (*n* = 36). In general, soil pH positively correlates with genes in the functions enriched in M1, and negatively correlated with that in M2. Note that associations between environmental variables and genes in specific functions are detailed in Supplementary Figs. 12–18. Source data are provided as a Source Data file.

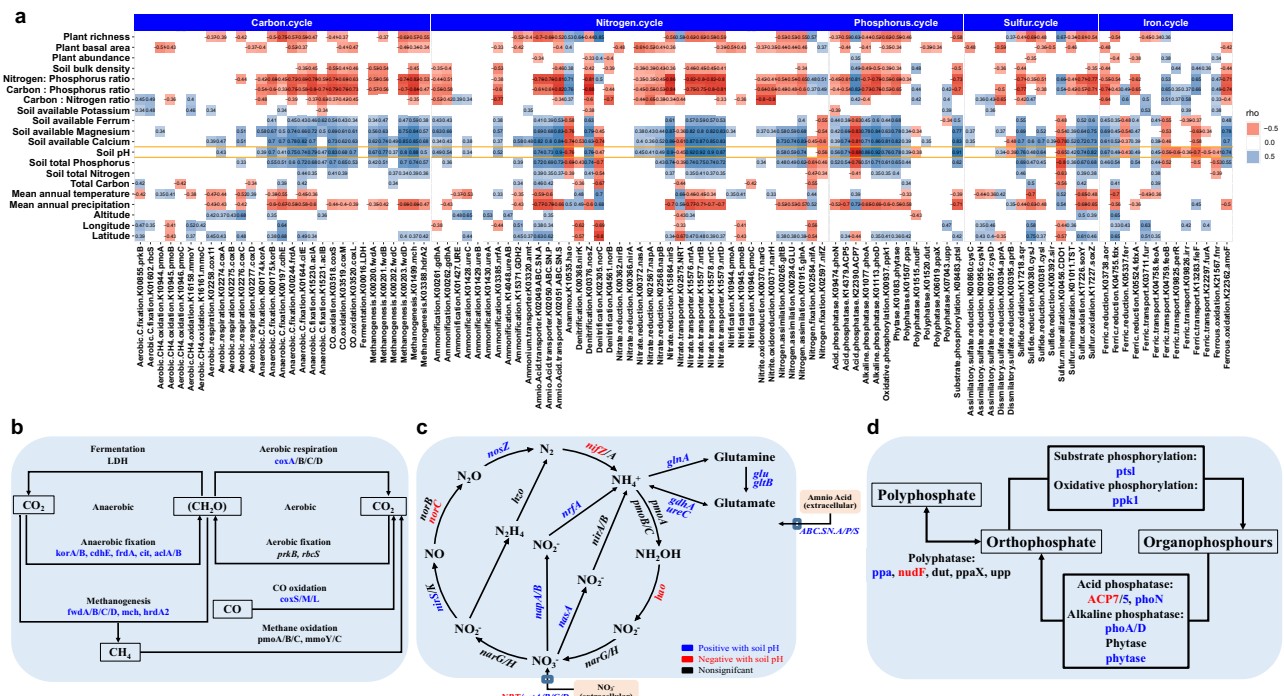

**Fig. 5 | Genes involving biogeochemical cycle associated with environmental variables. a** Heatmap showing correlations between environmental variables and genes involving biogeochemical cycling of carbon, nitrogen, phosphorus, sulfur and iron (*n* = 36 samples). **b** Seventeen carbon cycle genes positively correlate with soil pH. **c** Nitrogen cycle genes are positively or negatively correlated with soil pH.

**d** Eight phosphorus cycle genes positively correlate with soil pH, as compared to that two phosphorus cycle genes negatively correlate with soil pH. The diagrams for sulfur and iron cycles can be found in Supplementary Fig. 18. Source data are provided as a Source Data file.

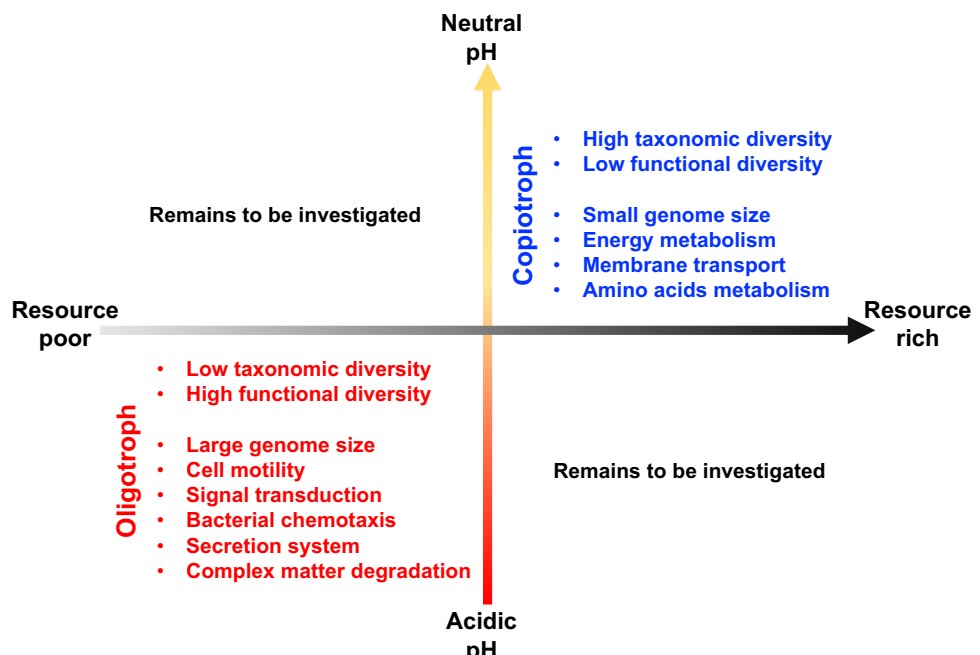

**Fig. 6 | Conceptual model on the adaptive strategy of soil microbiome along a latitudinal pH gradient across 12 forests.** Oligotrophs characterized by larger genome are adapted to the acidic, resource-poor soil, with an enrichment on functions of cell motility, bacterial chemotaxis, secretion system, signal transduction and complex matter degradation. The copiotrophs characterized by smaller genome are adapted to the neutral, resource-rich soil, with an enrichment on the functions of energy metabolism, membrane transport and amino acid metabolism. Note in our study system the resource availability was poor in acid pH soil, and rich in neutral pH soil. However, resources availability is known to be low in some neutral pH soil and high in some acid pH soil and microbial functional traits and adaptive strategies in these two conditions remain unresolved.

possibility of using genome size to study the ecological processes of community assembly and ecological networking[46]. Microbial community biodiversity occupies a central role in ecology, as the driver of ecosystem function and stability, and as the force behind emergent properties of microbial communities[47,48]. Given the central role of microbial communities in ecosystems, we hope that our research will stimulate additional research and debates over the characterization of microbial communities and the integration of taxonomic and functional diversity, which can deepen our understanding of processes of community assembly and ecosystem function.

## Methods

### Sites and soil sampling

This study sampled soils from 12 forest sites covering temperate, subtropical, and tropical climatic zones (Fig. 1a). Soil samples in the forest of Genhe (GH) were collected in long-term observation plots. In the other 11 forests, the Chinese Forest Biodiversity Monitoring Network (CForBio) had established a permanent plot (15–25 ha) consisting of hundreds of 20 m × 20 m quadrats (375–625 quadrats)[49]. Three quadrats in each site were randomly selected for soil sampling in this study. In each quadrat, ten soil cores (10 cm in depth, 5 cm in diameter) were evenly collected and mixed to make one composite sample, resulting in a total of 36 soil samples (3 quadrats × 12 forests). The soil samples were immediately sealed in plastic bags and transported to laboratory on ice. After removing stones and plant debris, fresh soil samples were sieved to 2 mm mesh size. Then, each sample was divided into two subsamples. One subsample was stored at −80 °C until DNA extraction, and another subsample was air-dried and used to measure soil abiotic properties.

### Soil properties and climatic factors

Soil pH was measured in a 1:2.5 soil/water suspension. Soil total carbon (TC) and total nitrogen (TN) were measured with an Elementar Vario EL III (Elementar Analysensysteme GmbH, Germany), and soil total phosphorus (TP) was measured by an Inductively Coupled Plasma-Atomic Emission Spectrometer (iCAP 6300, Thermo Jarrell Ash Co.). Soil available calcium, magnesium, iron and potassium were extracted by Mehlich-III solution and measured with Atomic Emission-Inductively Coupled Plasma (ICP-AES, Avio 500, PerkinElmer)[50]. Latitude, longitude, soil bulk density, soil available cations and plant data (abundance, richness and basal area) of the study sites were provided by plot founders and the CForBio organization. Latitude and longitude were recorded by GPS device. Soil bulk density was calculated as the dry weight of soil divided by its volume. Plant community data were surveyed by manual. The mean annual temperature (MAT) and mean annual precipitation (MAP) were obtained from the WorldClim database (www.worldclim.org) with a resolution of 2.5 min[51].

### DNA extraction and sequencing

The soil DNA of each sample was extracted by using the PowerSoil DNA isolation Kit (MoBio, Carlsbad, CA, USA). The concentration and purity of each DNA sample was determined by a Qubit 2.0 Flurometer (Life Technologies, CA, USA) and a NanoDrop2000 (Thermo Fisher Scientific, Waltham, MA, USA), respectively. The quality of DNA extracts was checked by electrophoresis in 1% agarose gels. Shotgun metagenomic sequencing of each DNA sample was sequenced on an Illumina HiSeq 2500 instrument (San Diego, CA, USA) with the PE125 run mode at Novogene, Inc. Beijing, China. Before sequencing, the DNA was fragmented into small components, and paired-end libraries were constructed using NEXTflexTM Rapid DNA-Seq (Bioo Scientific, Austin, TX, USA). Adapters containing the full complement of sequencing primer hybridization sites were ligated to the blunt end of DNA fragments. The raw sequence data underwent the following quality control process: paired reads were discarded if either read contained adapter contamination, or if more than 10% of bases were uncertain in either read, or if the proportion of low quality (Phred quality <5) bases was over 50% in either read. After data quality control, we had 509.64 GB

data of clean sequence for the 36 samples with average of 14 GB per sample.

From the same DNA, the bacterial 16S rRNA gene was amplified with the primer pair, 515 F/806 R[52], in the following PCR amplification solution: 2.5 µl 10 × buffer, 1.5 mM MgSO$_4$, 200 µM of each dNTP, 0.75 µM of each primer, 0.5 U KOD-plus-Neo Polymerase (Toyobo Co., Ltd., Osaka, Japan), and ~10 ng template DNA. A 12-base pair barcode, which is unique for each sample, was on the forward primer 515 F. The thermal cycling consisted of an initial denaturation at 95 °C for 3 min, followed by 30 cycles of denaturation at 95 °C for 50 s, annealing at 56 °C for 1 min, and extension at 68 °C for 1 min, followed by a final extension at 68 °C for 10 min. The PCR products were purified using a gel purification kit (Axygen), and 50 ng of DNA from each sample was pooled and adjusted to 10 ng µl$^{-1}$. After adding sequencing adapter to the PCR products using an Illumina TruSeq DNA PCR-Free Library Preparation Kit (Illumina) following the manufacturer's instructions, the library was then sequenced on an Illumina MiSeq PE250 instrument (San Diego, CA, USA) at Chengdu Institute of Biology, Chinese Academy of Sciences.

### Metabarcoding analyses

Sequences of 16S rRNA gene amplicon were aligned to each sample according to the unique barcode at the 5′-end of the forward primer 515 F. Overall sequencing quality was evaluated using FastQC v0.11.5[53]. Forward and reverse reads were merged using the fastq_mergepairs command in USEARCH v8.0[33]. Primers were removed using cutadapt v1.9.1[54]. Quality control was carried out using the fastq_filter command (-fastq_maxee 1.0-fastq_minlen 200) in USEARCH[33]. Chimeras were detected and removed in USEARCH with -uchime_ref command[33]. High-quality non-chimeric sequences were subjected to de-replication and de-singleton, and then clustered into operational taxonomic units (OTUs) at a 97% sequence similarity level using the cluster_otus command in USEARCH[33]. OTUs were identified by a BLAST search of the most abundant sequence representing that OTU against the Greengenes database for bacteria[55].

### Metagenomic analyses

To analyze microbial genomic traits based on metagenomes, the clean sequences returned from the sequencing facility were assembled into contigs for each sample using MEGAHIT v1.2.9 with --k-list 21, 29, 39, 59, 79, 99, 119, 141[56]. To focus on bacteria[13], we detected and removed potential eukaryotic contigs using EukRep v0.6.7[57], and detected and removed potential viral contigs using VIBRANT v1.2.1[58] and BBMap v39.01 (https://github.com/BioInfoTools/BBMap/). Average genome size of each sample was evaluated by using MicrobeCensus pipeline[29]. In brief, the pipeline evaluates genome size based on sequence coverage of 30 core single-copy genes which were found to universally present in bacteria and archaea. These essential genes can be sequenced at a higher rate, i.e., higher coverage, when average genome size of community is smaller, as these genes should make up a higher fraction in a small genome size. Overall, higher sequence coverage of these 30 genes indicates smaller average genome size. Average 16S rRNA gene copy number was evaluated by following method from ref. 15. In brief, average 16S rRNA gene copy number was estimated by dividing coverage of 16S rRNA gene by number of genomes in a metagenome. GC% was calculated by using Quast v5.2.0[30]. Minimal doubling time was estimated by gRodon v2.3.0[27] which analyzed codon usage bias and calculated minimal doubling time based on the tight relationship between codon usage bias and bacterial maximum growth rate[59]. GC% was calculated based on all assembled contigs, while average genome size, average 16S rRNA gene copy number and minimal doubling time were calculated based on contigs >500 bp as length could affect the accuracy of estimation methods.

We also calculated bacterial average genome size, protein counts and GC% by relating annotated 16S rRNA gene sequence to the Genome Taxonomy Database (GTDB)[31]. Metadata file and FASTA file of 16S rRNA gene sequences of bacterial representative genomes were download from GTDB (https://gtdb.ecogenomic.org/downloads). The FASTA file was made as a reference database by using makeblastdb v2.13.0[60]. Then, we annotated our 16S rRNA gene amplicon against the reference database by using BLAST v2.13.0[60] to get genome accession ID. Data of genome size, protein counts and GC% were acquired by matching accession ID to the metadata file. Bacterial average genome size, protein counts and CG% of each sample were calculated with weighing reads counts.

To analyze specific genes, all assembled contigs were combined. Then, the redundans pipeline (https://github.com/lpryszcz/redundans) was used to detect and selectively remove redundant contigs with the settings --overlap 80 and identity 0.90[61]. Prodigal v2.6.3 was used to predict protein-coding genes[62], resulting in 20,817,601 genes. These gene sequences were clustered by using CD-HIT v4.8.1 with sequence identity threshold 0.95 and alignment coverage 0.9[63], resulting in a non-redundant gene catalog with 20,038,815 genes. Salmon v1.6.0 was used to estimate gene abundance in each sample by mapping raw reads to the non-redundant gene catalog, considering reads counts and gene length[64]. The non-redundant gene catalog was annotated by using EggNOG-mapper tool through against eggNOG database (http://eggnog5.embl.de)[65] that integrated databases of Clusters of Orthologous Groups of proteins (COGs)[66], Kyoto Encyclopedia of Genes and Genomes (KEGG) Orthology (KO)[34] and the carbohydrate-active enzymes (CAZy)[67]. In addition, antibiotic resistance (AR) genes were annotated against Resfams database[37] by using DIAMOND tool (https://github.com/bbuchfink/diamond)[68]. Besides functional annotations, taxonomic classification based on sequences of metagenome was done by Kaiju (https://kaiju.binf.ku.dk/), which is a program for fast taxonomic classification based on sequence comparison to a reference database of microbial proteins[69].

### Statistical analyses

To check if soil pH drives bacterial taxonomic and function distribution along the pH gradient across 12 forests, principal coordinate (Pco) analysis in the "*stats*" package was used to visualize bacterial taxonomic and functional composition, and envfit analysis in the "*vegan*" package was used to analyzed the relationship between bacterial taxonomic and functional composition and plant, soil, climate and geography variables[70]. The linear regression model in the "*stats*" package was used to test if the relative abundance of *Acidobacteria* decreased with increasing soil pH.

To analyze the distribution pattern of bacterial genome traits across the pH gradient, the linear regression model in the "*stats*" package was used to analyze the relationships between soil pH and average genome size, average 16S rRNA gene copy number, GC%, and minimal doubling time. Relationships between these genome traits and plant, soil, climate and geographic variables were tested by Spearman's correlation analysis, using corr.test function in the "*psych*" package[71].

To check previous findings that bacterial taxonomic diversity of 16S rRNA gene amplicon increases from acid to neutral pH (Supplementary Table 1), we used our 16S rRNA gene amplicon dataset to calculate richness (S.16S) and the Shannon diversity index (H'.16 S) using the diversity function in the "*vegan*" package[70]. These parameters were subsequently regressed against soil pH. To explore if bacterial functional diversity decreases from acid to neutral pH, for KEGG-annotated metagenomic results we calculated the richness (S.KO) and Shannon diversity index (H'.KO), which were subsequently regressed against soil pH. Similarly, regression was carried out to explore the relationship between soil pH and the richness and Shannon diversity of specific functions of carbohydrate-active enzymes (annotated by CAZy databse) and antibiotic resistance genes (annotated by Resfams database). Relationships between bacterial taxonomic

diversity and functional diversity were analyzed by a linear regression model in the "*stats*" package. Furthermore, to test if functional diversity positively correlated with average genome size, regression was carried out between functional diversity indices and average genome size.

To explore the distribution of abundance of functional genes along the pH gradient, we employed co-occurrence network analysis to demonstrate the associations among all KEGG Orthology terms (KOs). First, only KOs that occurred in >18 of all 36 samples were kept for network analysis, yielding 7717 KOs. Then, co-occurrence among annotated KOs was detected using pairwise Spearman's correlations in the "*psych*" package[71]; the correlations were used to construct co-occurrence by the "*igraph*" package[72] if its Spearman's rho >0.6 and FDR $P < 0.05$. The co-occurrence network was composed of 7481 vertices (KOs) and 1,359,406 edges, and clustered into two big modules: module 1 (M1) containing 2,777 vertices and module 2 (M2) containing 4309 vertices. Regression analysis in the "*nlme*" package[73] was used to test the relationship between soil pH and vertices in the two biggest modules, M1 and M2.

To explore functional difference of KOs between these two modules, from these KOs we reconstructed KEGG pathway maps using KEGG Mapper (https://www.kegg.jp/kegg/mapper/). Then, KEGG pathways with significantly different KO counts between M1 and M2 (FDR < 0.05 and |logFC | >= 0.5) were explored using differential analysis implemented in the "*edgeR*" package[74].

In addition, Mantel analysis in the "*linkET*" package[75] was used to analyse the relationships between genome traits, diversity indices and taxonomic and functional compositions. Spearman's correlation analysis in the "*psych*" package[71] was further used to test the relationship between relative abundance of functional genes and plant, soil, climate and geographic variables. Threshold indicator taxa analysis (TITAN) was used to explore pH preference of KO, using the "*TITAN2*" package[76]. Data were log- or sqrt-transformed to meet homogeneity of variance when necessary. All statistical analyses were conducted on R statistical programming language version 4.2.1 if else stated[77].

## Reporting summary

Further information on research design is available in the Nature Portfolio Reporting Summary linked to this article.

## Data availability

All metabarcoding and metagenomics sequences in this study have been deposited in the National Center for Biotechnology Information database (PRJNA986291). The WorldClim database is available through https://www.worldclim.org. Source data are provided with this paper.

## Code availability

The source code implementing the analyses in this manuscript is available on Github (https://github.com/FunCongWang/CForBio.metagenome) or Zenodo[78] (https://zenodo.org/records/10017052).

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

## Acknowledgements

This work was supported by the Strategic Priority Research Program of the Chinese Academy of Sciences [XDA28030401 (C.G.), XDB31030000 (L.G.)], National Natural Science Foundation of China [No. 32322053 (C.G.), 32101286 (C.W.), 32170129 (C.G.)]. We thank support from the Chinese Forest Biodiversity Monitoring Network and Chinese Soil Microbial Observation Network. We also thank support from the Inner Mongolia Daxing'anling Forest Ecosystem Research Station, Liangshui Experimental Forest Farm, the Research Station of Changbai Mountain Forest Ecosystem, the Beijing Forest Experimental Station, the Baotianman National Nature Reserve, the East China Normal University Tiantong National Forest Ecosystem Observation and Research Station, the Badagong Mountain National Nature Reserve, the Zhejiang Qianjiangyuan Forest Biodiversity National Observation and Research Station, the Heishiding Nature Reserve, the Dinghu Mountain National Nature Reserve, the Nonggang National Nature Reserve, and the National Forest Ecosystem Research Station at Xishuangbanna. We thank Mohammad Bahram for sharing some data of Bahram et al.[7]. We are grateful to Min Cao, Liang Chen, Xiao-Yong Chen, Xiao-Bao Deng, Xiao-Jun Du, Zhan-Qing Hao, Ming-Xi Jiang, Guang-Ze Jin, Bu-Hang Li, Xian-Kun Li, Xing-Chun Li, Ke-Ping Ma, Wei-Guo Sang, Xiang Sun, Yong-Long Wang, Wan-Hui Ye, Ming-Jian Yu, and Xiao-Liang Zhang for help in field sampling. We thank Kabir G. Peay and Noah Fierer for feedback on the manuscript, and Qi Wu for help in bioinformatics. We thank all members of Gao Lab for their discussion and communications.

## Author contributions

C.W., L.G., J.T., and C.G. conceived of and wrote the manuscript. C.G., N.J., Y.Z., and L.G. coordinated the field sampling and molecular analysis. C.W., Q.Y., and C.G. performed the bioinformatics and statistical analyses.

## Competing interests

The authors declare no competing interests.
