## [Peer Review File · Nature Communications]

Bacterial genome size and gene functional diversity negatively correlate with taxonomic diversity along a pH gradientReviewer #1 (Remarks to the Author):

In this study, the authors describe relationships between taxonomic and functional aspects of diversity and genome size across a pH gradient. They find taxonomic and functional diversity to be inversely correlated across the pH gradient, an observation they attribute to differences in genome size between taxa from bacterial communities found in high versus low pH soils. By incorporating the analysis of functional genes across the gradient, the study builds a case for life history strategies that might characterize taxa adapted to the different ends of the pH gradient. These life history strategies are used to explain the apparently unintuitive relationships observed between taxonomic and functional diversity and genome size across this gradient.

While the study is supported by a thorough and sound data collection effort and extensive analyses, there are a number of problematic points that in my opinion do not make this piece suitable for publication and therefore I recommend rejection of this work. Generally, a decoupling between functional and taxonomic diversity has already been reported in a number of previous studies (most of them cited in this work), and relationships between aspects of bacterial diversity and pH are often system-dependent (as shown by the compilation in Suppl. Table 1). I hope my comments below help the authors improve their manuscript:

1- The generalizability of the findings is unclear: The authors base their hypotheses on the assumption that pH gradients underpin gradients in resource availability (L46-47, unsupported by any references). While it is true that for nutrient elements like phosphorus low pH reduces availability in soil (but not total soil P content as measured here), this is not a universal situation for many other resources. Across this study, samples with high total N, C, P, and Ca correspond to samples with higher pH, which is clearly not generalizable to other systems (for example, across Australia high pH soils have low total C, P, and N – an opposite trend to this work - <https://journals.asm.org/doi/full/10.1128/mbio.01718-20#tabS1>). The conclusions regarding oligotrophic versus copiotrophic life history strategies that sustain the patterns in genome size of the study are therefore not generalizable with the data presented. The correlations between pH and genome size observed here might therefore be a particularity of this study and not something that challenges prior tenets. The study would need to include data from diverse soil pH gradients in order to establish general associations between genome size, life history strategies, and aspects of diversity.

2- Poor grammar, vocabulary, and structure: The text is deficient in many aspects of English writing. An example of poor writing is L72-74, among many others. Another aspect is for example the notation to indicate statistical significance: L112 shows "P < 0.001", L113 shows "P = 3.925 e-06", L185 shows "P < e-16". It is somehow striking that none of the signing authors has paid more attention to this, and that any potential native English speaker among them has read and corrected the many issues with English in the text.

3- The decoupling between taxonomic and functional diversity is presented as a novel finding, but there is no mechanistic insight into why this decoupling happens. The authors could easily check if the most abundant taxa are also those with larger genomes in acidic soils, and if these genomes also have a higher diversity of functional genes. The authors can do that by matching the 16S rRNA gene sequences from their amplicon-based effort to the genome taxonomy database (GTDB), and annotate those individual genomes. This would make the article less speculative and make the discussion more compelling. They could also address the fact that genome size and GC content are negatively correlated in their communities using the same "genome-specific" approach, given these traits are universally positively correlated in the bacterial world.

4- The authors could establish other mechanistic links to pH adaptation to support the generalizability of their findings by looking for the prevalence of genes associated with pH preference/tolerance that have been identified in diverse studies previously (including the Malik et al. study cited in this work).

Reviewer #2 (Remarks to the Author):

Review: NCOMMS-23-24916-T

General Report:

What are the noteworthy results?

This manuscript reports the negative correlation between taxonomic and functional diversity along a pH gradient, which is tied to bacterial genome size. Furthermore, they demonstrate environmental selection for relevant functional genes. Both of these findings are highly relevant for current and future interests in environmental microbiology, and the framework by which they conducted the study will also be highly useful for conducting this type of analysis along other environmental gradients.

Will the work be of significance to the field and related fields? How does it compare to the established literature? If the work is not original, please provide relevant references.

This work is significant to the field as it provided strong evidence of the suspected antithetical relationship between microbial taxonomic and functional diversity and provides a strong framework by which to evaluate the relationship. In addition, it advances our understanding of the selectivity of the individual environments studied and further investigates the consequences of the selectivity by addressing key biochemical pathways. In effect, this paper is a natural continuation of work that has been previously published, but furthers the analysis in an elegant and useful manner.

Does the work support the conclusions and claims, or is additional evidence needed?

The work thoroughly supports the conclusions and claims.

Are there any flaws in the data analysis, interpretation and conclusions? Do these prohibit publication or require revision?

I see no flaws in the analysis and appreciate the authors' attention to detail regarding the specific bioinformatic pipelines used. The thoroughness of the methods ensures reproducibility of the results, and also allows the manuscript to also serve as a reference for the specific analysis framework the authors' developed.

Is the methodology sound? Does the work meet the expected standards in your field?

Yes, I believe the methodology is sound and surpasses the standards in the field. As mentioned above, I found the thoroughness of the methods section to be enlightening and appreciate the authors' effort.

Is there enough detail provided in the methods for the work to be reproduced?

Yes, the level of detail included in the methods of this manuscript will allow for reproduction/replication of the analysis pipeline.

Specific comments:

As detailed above, I was thoroughly impressed with this manuscript from both the results-oriented standpoint, and the usefulness of the analysis pipeline. I have no major issues with the manuscript as is, and feel the work is highly relevant and likely to advance the use of these types of analysis in the future.

However, I do suggest the authors revisit the manuscript with a Native-English Editor as there are some deficiencies in grammar and spelling that need attention prior to publication. These issues did not hinder the comprehension of the manuscript except in a few cases, but are still vital for the suitability of the manuscript for publication.

The figures for this manuscript are well-designed, but it is my suggestion to revise the color-scheme used for at least figure 5. The lime/light green color was hard to read against the blue background, even when the full-size figure was observed on a high-resolution computer monitor. Please change the color-scheme for this and figure 4 which both predominantly feature the contrast of lime-green and dark blue.

Lastly, please check capitalization and punctuation in the references section as there are some inconsistencies in sentence case.

Reviewer #1 (Remarks to the Author):

In this study, the authors describe relationships between taxonomic and functional aspects of diversity and genome size across a pH gradient. They find taxonomic and functional diversity to be inversely correlated across the pH gradient, an observation they attribute to differences in genome size between taxa from bacterial communities found in high versus low pH soils. By incorporating the analysis of functional genes across the gradient, the study builds a case for life history strategies that might characterize taxa adapted to the different ends of the pH gradient. These life history strategies are used to explain the apparently unintuitive relationships observed between taxonomic and functional diversity and genome size across this gradient.

Response: We appreciate the reviewer's interest and careful reading on our work.

While the study is supported by a thorough and sound data collection effort and extensive analyses, there are a number of problematic points that in my opinion do not make this piece suitable for publication and therefore I recommend rejection of this work. Generally, a decoupling between functional and taxonomic diversity has already been reported in a number of previous studies (most of them cited in this work), and relationships between aspects of bacterial diversity and pH are often system-dependent (as shown by the compilation in Suppl. Table 1). I hope my comments below help the authors improve their manuscript:

Response: We thank the reviewer for the high appraisal on our data collection and analysis efforts. We also thank the reviewer for the insightful comments that helped us to prepare a much-improved manuscript.

Regarding the decoupling between functional and taxonomic diversity, we respectfully do not agree with the reviewer that this area *'has been reported in a number of previous studies'*. Our literature search effort found that only one recent study reported a decoupling between bacterial functional and taxonomic diversity. Specifically, Kerfahi, et al. ¹ found that diversity of functional genes was negatively correlated with soil pH along a gradient of increasing altitude on Mt. Norikura, Japan. Thus, to our best knowledge, the decoupling between functional and taxonomic diversity remains is not widely documented, especially at regional scale.

Regarding the relationships between aspects of bacterial diversity and pH, we respectfully do not agree with the reviewer that this area 'are often system-dependent'. As shown in the Supplementary Table 1, a compilation of previous studies showed that for 37 studies testing the relationship between soil pH and bacterial taxonomic diversity, 34 of them reported a positive relationship between soil pH and bacterial taxonomic diversity (peak at neutral pH), ranging from local, regional to global scale studies. For the three studies that failed to detect a positive correlation between pH and bacterial taxonomic diversity, the pH gradient was not well represented as the pH ranged either 3.6-4.5, or 5-9, or 7-10.5 in these studies. Thus, in our opinion, previous documentation of the relationships between bacterial taxonomic diversity and pH was consistent and largely not system dependent.

1- The generalizability of the findings is unclear: The authors base their hypotheses on the assumption that pH gradients underpin gradients in resource availability (L46-47, unsupported by any references). While it is true that for nutrient elements like phosphorus low pH reduces availability in soil (but not total soil P content as measured here), this is not a universal situation for many other resources. Across this study, samples with high total N, C, P, and Ca correspond to samples with higher pH, which is clearly not generalizable to other systems (for example, across Australia high pH soils have low total C, P, and N – an opposite trend to this work - <https://journals.asm.org/doi/full/10.1128/mbio.01718-20#tabS1>). The conclusions regarding oligotrophic versus copiotrophic life history strategies that sustain the patterns in genome size of the study are therefore not generalizable with the data presented. The correlations between pH and genome size observed here might therefore be a particularity of this study and not something that challenges prior tenets. The study would need to include data from diverse soil pH gradients in order to establish general associations between genome size, life history strategies, and aspects of diversity.

Response: First, for our hypothesis, we provided reference that could support the assumption that pH gradient underpin gradient in resource availability. Specifically, Jordan ², Vitousek and Sanford ³ and a number of other previous studies have found that acid soil pH in tropical forests is associated with poor resource availability, whereas neutral soil pH in temperate forests is

associated with rich resource availability. In line with this assumption, as pointed out by the reviewer, our data collected along a latitudinal gradient of 12 forests also showed that the acid soil pH in tropical forests are associated with poor resource availability, whereas neutral soil pH in temperate forests are associated with rich resource availability (Supplementary Fig. 8). This condition represents the top-right and bottom-left quadrats of our framework, as shown in Supplementary Fig. 18.

On the other hand, we agree with the reviewer that high soil pH can be low in resources of total C, N and P, and thereby limiting soil bacterial community and function as shown by the aforementioned Oliverio, et al. ⁴ paper. Although this condition is not covered by our sampling efforts in various forest ecosystems in China, it is included in our framework in Supplementary Fig. 18.

Supplementary Fig. 8. Bacterial taxonomic and functional compositions in association with biotic and abiotic variables. The heatmap in left-bottom triangle showing the intercorrelations among microbial genomic traits, microbial diversity indices, plant, soil and geographical variables as detected by Spearman's correlation analysis, with a color gradient (red to blue) and box size denoting Spearman's correlation coefficients (rho). The significance of Spearman's correlation is denoted by * $P < 0.05$, ** $P < 0.01$, *** $P < 0.001$. The curved lines in the top-right triangle show the association of bacterial taxonomic (the right-top dot) and functional (the

right-bottom dot) compositions with biotic and abiotic variables, as detected by Mantel tests. The width of curved lines corresponds to the r statistic for Mantel test, and the color of curved lines denotes the statistical significance. AGS: average genome size; ACN: average 16S rRNA gene copy number; Doubling time: minimal doubling time; MAP: mean annual precipitation; MAT: mean annual temperature; TC: total carbon; TN: total nitrogen; TP: total phosphorus; ACa: available calcium; AMg: available magnesium; AFe: available iron; AK: available potassium; C_N: carbon nitrogen ratio; C_P: carbon phosphorus ratio; N_P: nitrogen phosphorus ratio. $n = 36$.

Supplementary Fig. 19. Conceptual model on the adaptive strategy of soil microbiome along a latitudinal pH gradient across 12 forests. Oligotrophs characterized by larger genome and lower GC% are adapted to the acidic, resource poor soil, with an enrichment on functions of cell motility, bacterial chemotaxis, secretion system, signal transduction and complex matter degradation. The copiotrophs characterized by smaller genome and higher GC% are adapted to the neutral, resource rich soil, with an enrichment on the functions of energy metabolism, membrane transport and amino acid metabolism. Note in our study system the resource availability was poor in acid pH soil, and rich in neutral pH soil. However, resources availability is known to be low in some neutral pH soil and high in some acid pH soil and microbial functional traits and adaptive strategies in these two conditions remain unresolved.

Added text in lines 286-295: We found that variation in genome size influenced the relationship between taxonomic and functional diversity in systems where resource availability was poor, as in acid pH soil, or rich, as in neutral pH soil. However, it is known that resource availability can be low in neutral pH soil and high in acid pH soil⁵, showing that studies combining taxonomic diversity with functional diversity should be extended to additional ecosystems and additional microbial communities. Thus, our understanding of the structure and function of soil microbiomes would be advanced by a thorough investigation of the functional traits, diversity and adaptive strategy of microbiomes that represent the four quadrants defined by the axes of resource and stress (Supplementary Fig. 19).

To address the reviewer's concern that the '*correlations between pH and genome size observed here might therefore be a particularity of this study and not something that challenges prior tenets*', we re-analyzed the global topsoil metagenome data of Bahram, et al. ⁵. As shown in the following figures (Supplementary Fig. 4 in the manuscript), the reanalysis of Bahram, et al. ⁵ found that soil pH significantly negatively correlated with bacterial average genome size that detected by either metagenome using MicrobeCensus or by 16S rRNA amplicon referencing GTDB. Thus, the negative correlation between pH and genome size is not likely limited to our study in forests in China, but more likely generalizable to large scale and various ecosystem types.

Supplementary Fig. 4. Re-analysis of Bahram et al 2018 showing associations of bacterial average genome size with soil pH. a Soil pH negatively correlated with average genome size as detected by shotgun metagenome using MicrobeCensus pipeline. **b** Soil pH negatively correlated with average genome size as detected by 16S rRNA amplicon referencing Genome Taxonomy Database (GTDB). n = 134.

Added text in lines 154-159: The generality of our finding that bacterial genome size decreased from acid to neutral pH in China was tested by re-analyzing the previously published global dataset of Bahram, et al. ⁶. Using both the MicrobeCensus annotation of metagenomes and the matching of 16S rRNA amplicons with GTDB, our reanalysis of Bahram, et al. ⁶ again found that bacterial average genome size was negatively correlated with soil pH (Supplementary Fig. 4).

2- Poor grammar, vocabulary, and structure: The text is deficient in many aspects of English writing. An example of poor writing is L72-74, among many others. Another aspect is for example the notation to indicate statistical significance: L112 shows “ $P \ll 0.001$ ”, L113 shows “ $P = 3.925 \times 10^{-6}$ ”, L185 shows “ $P < 10^{-16}$ ”. It is somehow striking that none of the signing authors has paid more attention to this, and that any potential native English speaker among them has read and corrected the many issues with English in the text.

Response: We appreciate this comment on English usage and have thoroughly edited our entire manuscript for English usage.

Original text in lines 72- 74: Different taxa with large genome can harbor more versatile gene, to the end even a relatively low diversity of large genome taxa can support a high diversity of functional genes.

Revised: Lines 76-80: ...small genomes are likely to be dominated by indispensable, core genes^{20,21}, whereas large genomes can harbor the core genes as well as genes of diverse function⁹. As a result, a relatively low diversity community of large genome taxa can support a higher diversity of functional genes than a high diversity community of small genome taxa.

3- The decoupling between taxonomic and functional diversity is presented as a novel finding, but there is no mechanistic insight into why this decoupling happens. The authors could easily check if the most abundant taxa are also those with larger genomes in acidic soils, and if these genomes also have a higher diversity of functional genes. The authors can do that by matching the 16S rRNA gene sequences from their amplicon-based effort to the genome taxonomy database (GTDB), and annotate those individual genomes. This would make the article less speculative and make the discussion more compelling. They could also address the fact that genome size and GC content are negatively correlated in their communities using the same “genome-specific” approach, given these traits are universally positively correlated in the bacterial world.

Response: We thank the reviewer for pointing out that changes in genome size of the abundant taxa can be a mechanistic insight on our novel finding that bacterial taxonomic diversity decouples from functional diversity. Based on our previous discussion on the role of average genome size

on the decoupling between taxonomic and functional diversity, we now added more analysis and discussion to make this area clearer.

1) Our analysis of community composition showed that the changes of 16S-rRNA bacterial community composition along increasing soil pH were dominated by the gain of DA101 (*Verrucomicrobia*) and the loss of *Ca. Solibacter* (*Acidobacteria*), *Ca. Xiphinematobacter* (*Verrucomicrobia*), *Ca. Koribacter* (*Acidobacteria*), *Burkholderia* (*Proteobacteria*), and *Salinispora* (*Proteobacteria*). A search using GTDB, NCBI and published literature found that the mean genome size of DA101 is 2.8 Mb ⁶, whereas the average genome size is 5.5-8.5 Mb for three bacteria genera that show reduced relative abundance. The exception to this reduction is *Ca. Xiphinemetobacter* with only a 0.9 Mb genome size, but its abundance is too low to affect the trend. Thus, the change of bacterial community composition along increasing soil pH was associated with the gain of some small genome size taxa, and loss of several large genome taxa. We presented this result in Supplementary Fig. 3.

Supplementary Fig. 3. Associations of relative abundance of top 10 bacterial genera with soil pH. The average genome size for each genus is searched from GTDB, NCBI or published literature ⁴⁹. n = 36.

2) Our matching of 16S-rRNA bacterial community composition with bacterial genomes from GTDB showed that average genome size was negatively correlated with soil pH (Supplementary

Fig. 2), a result consistent with our metagenome-based analysis (Fig. 2a). Our GTDB based analysis also found that the average genome size was strongly positively correlated with coding gene number per genome. These results are presented in Supplementary Fig. 2.

Supplementary Fig. 2. Genomic traits along a pH gradient. Bacterial average genome size and protein counts per genome are calculated by annotating 16S rRNA gene sequence to Genome Taxonomy Database (GTDB). **a** Bacterial average genome size decreased as pH changed from acidic to neutral. **b** Bacterial protein counts per genome significantly, positively correlate with average genome size. $n = 36$.

Added text in lines 140-153: In order to challenge our finding that bacterial genome size, calculated from the MicrobeCensus pipeline²⁸, is larger in acidic than in neutral pH environments (Fig. 2a), we additionally calculated the community-weighted genome size by referencing the 16S rRNA gene amplicon dataset against the Genome Taxonomy Database (GTDB)³⁰. The results again showed that the average genome size of the bacterial community was significantly, negatively correlated with soil pH (Supplementary Fig. 2). The decrease of bacterial genome size was largely caused by changes with increasing soil pH of the most abundant bacterial taxa (Supplementary Fig. 3), including the gain of a genus with a small genome, DA101 (2.80 Mb), and the loss of large-genome taxa of *Ca. Solibacter* (5.52 Mb), *Ca. Koribacter* (5.65 Mb), *Burkholderia* (8.57 Mb), and *Salinispora* (5.56 Mb)³⁰ (Supplementary Fig. 3). One of the declining taxa, *Xiphinematobacter*,

harbors a very small genome (0.91 Mb, likely related to its parasitism of nematodes³¹), but was far less abundant than DA101 and too rare to affect the trend (Supplementary Fig. 3).

Interestingly, although our metagenome-based analysis showed that bacterial average genome size was negatively correlated with GC%, our analysis using the 16S rRNA-GTDB method found that bacterial community-weighted genome size was not significantly correlated with GC% (Supplementary Fig. 5). Similarly, our re-analysis on the global data of Bahram, et al.⁵ showed that metagenome-based bacterial average genome size was negatively correlated with GC%, whereas the 16S rRNA-GTDB based bacterial community-weighted genome size was significantly positively correlated with GC% (Supplementary Fig. 6). Given the inconsistent results between the genomic-based method and the metagenomic-based method, we prefer to not give a mechanistic discussion over this piece of results. On the other hand, our results suggest that future research is in need to focus on the cause and consequence of the different results on the microbial traits detected from the genomic-based and metagenomic-based methods. We presented these results in Supplementary Figs. 5-6.

Supplementary Fig. 5. Relationships between bacterial average genome size and GC content (GC%). a-b Metagenome-based analysis a showed that bacterial average genome size was negatively correlated with GC%, while the 16S rRNA-GTDB method b found that bacterial average genome size was not significantly correlated with GC%. n = 36.

Supplementary Fig. 6. Relationships between bacterial average genome size and GC content (GC%). a-b Re-analysis on the global data of Bahram et al 2018 showed that a metagenome-based bacterial average genome size was negatively correlated with GC%, whereas b 16S rRNA-GTDB based bacterial average genome size was significantly positively correlated with GC%. n = 134.

Added text in lines 160-166: Interestingly, for the datasets of both this study and that of Bahram, et al. ⁶, average genome size significantly, negatively correlated with GC% based on analysis of the metagenome (Supplementary Figs. 5-6); however, the average genome size positively correlated with GC% when based on the matching of 16S rRNA community with GTDB (Supplementary Figs. 5-6). This obvious, methodological bias should stimulate research on the cause and consequence of the different results in microbial traits detected from the genomic-based and metagenomic-based methods.

4- The authors could establish other mechanistic links to pH adaptation to support the generalizability of their findings by looking for the prevalence of genes associated with pH preference/tolerance that have been identified in diverse studies previously (including the Malik et al. study cited in this work).

Response: We appreciate the reviewer's suggestion that we look for the prevalence of genes associated with pH. By comparing our study with those of Malik, et al. ⁷ and Ramoneda, et al. ⁸, we find that our suggestion of an oligotrophic lifestyle for acid pH-adapted microbes and copiotrophic lifestyle for neutral pH-adapted microbes is in line with the pH-gene associations reported by Malik, et al. ⁷ and Ramoneda, et al. ⁸. We provided this information in Supplementary

Table 2. The following table (Supplementary Table 2) showed the comparison of functions in association with pH in our study, Ramoneda, et al. ⁸ and Malik, et al. ⁷.

Added text in lines 232-235: These genes are involved with essential functions such as motility, bacterial secretion system, and xenobiotics biodegradation and metabolism, a finding largely in line with that of Ramoneda, et al. ³ (Supplementary Tables 2-3).

Added text in lines 242-245: Our detection of positive associations of pH with essential functions such as carbon metabolism and membrane transport is, again, largely consistent with the findings of Ramoneda, et al. ³ and Malik, et al. ²³ (Supplementary Tables 2-3).

Supplementary Table 2 Associations of certain microbial functions in this study*, in Ramoneda et al 2023 and in Malik et al 2018

This study (Wang et al)		Ramoneda et al 2023		Malik et al 2018	
Function (KEGG Level3 or gene)	pH association	Function (pfam)	pH association	Function	pH association
Carbon metabolism	(+)	Sugar metabolism	(+)	Carbon metabolism	(+)
Carbon fixation	(+)				
Glyoxylate and dicarboxylate metabolism	(+)				
Citrate cycle	(+)	Citrate transporter	(+)		
Methane metabolism	(+)				
ABC transporters	(+)	Transmembrane cation transporters/ Transmembrane anion transporter/Na ⁺ /H ⁺ antiporters	(+)	ABC transporters	(+)
Metabolism of other amino acids	(+)	Methionine metabolism	(+)	Biosynthesis of amino acids	(+)
Oxidative phosphorylation	(+)			Oxidative phosphorylation	(+)
Ribosome	(+)			Ribosome	(+)
Flagellar assembly	(-)	Motility	(-)		
Metabolism of terpenoids polyketides	(-)				
Xenobiotics biodegradation metabolism	(-)	Phenol degradation	(-)		
Glycan biosynthesis metabolism	(-)				
Bacterial secretion system	(-)	Type IV secretion system	(-)		
Bacterial chemotaxis	(-)				

Two component system	(-)	Transmembrane proteins	(-)		
Porphyrim metabolism	(-)				
Biosynthesis of siderophore	(-)				
Lipopolysaccharide biosynthesis	(-)				
MTHFD; methylenetetrahydrofolate dehydrogenase	(+)	Folate metabolism	(+)		
ATPase family AAA domain-containing protein 3A/B	(+)	ATPases/AAA_25	(+)		
uvrA/B,excinuclease ABC subunit B	(+)	UV damage repair endonuclease	(+)		
ACH1; acetyl-CoA hydrolase [EC:3.1.2.1]	(-)	Acetyl-CoA hydrolase/transferase	(-)		
pufC; photosynthetic reaction center cytochrome c subunit	(-)	Cytochrom C photosynthetic reaction center	(-)		
dexA; dextranase	(-)	Hydrolases of dextrans Sugar metabolism	(-)		
hipA/B	(-)	Kinase that inhibits tRNA synthase (antibiosis)	(-)		
hypF; hydrogenase maturation protein HypF	(-)	Hydrogenase maturation	(-)		
csxA; exo-1,4-beta-D-glucosaminidase [EC:3.2.1.165]	(-)	Glucosaminidase	(-)		
NAGLU; alpha-N-acetylglucosaminidase [EC:3.2.1.50]	(-)				
kdpA/B/C	(-)	K ⁺ transporter	(-)		
rhtB; homoserine/homoserine lactone efflux protein	(-)	Acylated homoserine lactone metabolism	(-)		
scrY; sucrose porin	(-)	Carbohydrate porin	(-)		
TC.FEV.OM3, tbpA, hemR, lbpA, hpuB, bhur, hugA, hmbR; hemoglobin/transferrin/ lactoferrin receptor protein	(-)	Transferrin dimerization domain	(-)		
plc; phospholipase C [EC:3.1.4.3]	(-)	Phospholipase C	(-)		
K14645; serine protease [EC:3.4.21.-]	(+)	Serine protease	(-)		
ccdA; cytochrome c-type biogenesis protein	(-)	Motif part of C type cytochrome	(-)		

napC; cytochrome c-type protein NapC	(+)				
ccmF; cytochrome c-type biogenesis protein CcmF	(+)				
nrfE; cytochrome c-type biogenesis protein NrfE	(+)				
perR; Fur family transcriptional regulator, peroxide stress response regulator	(-)	Stress response	(-)		
furA; Fur family transcriptional regulator, stress-responsive regulator	(-)				
universal stress protein A/G/F/E	(+)				
InuA_C_D_E, lin; lincosamide nucleotidyltransferase A/C/D/E	(+)	Nucleotidyltransferase	(-)		
pnp, PNPT1; polyribonucleotide nucleotidyltransferase [EC:2.7.7.8]	(+)				
				Purine metabolism	(+)
				RNA degradation	(-)
		Photolyase	(+)		
		Endonuclease/Exonuclease/phosphatase family	(+)		
		Lyase of methionine metabolism	(+)		
		Methyltransferases	(+)		
		Kinase that does AMPylation to proteins	(+)		
		Ca-dependent nuclease	(+)		
		Heme binding proteins	(+)		
		Gluc to Fruc 6-phosphate	(+)		
		Kinase that does AMPylation to proteins	(+)		
		Ca-dependent nuclease	(+)		
		Heme binding proteins	(+)		
		Ig like domains	(-)		
		Phosphatases in polysaccharide synthesis	(-)		
		Cytidylate kinase	(-)		
		Dehydrokinase	(-)		
		Helix-turn-helix domain	(-)		

		Lipid hydratase	(-)		
		Quinolones	(-)		
		Thyamin pirophosphate binding domain	(-)		
		Fe receptors	(-)		
		Lipid hydratase	(-)		

*Note a list of all KOs significantly associated with soil pH of our dataset as detected by threshold indicator analysis (TITAN) is provided in Supplementary Table 3.

Reviewer #2 (Remarks to the Author):

Review: NCOMMS-23-24916-T

General Report:

1. What are the noteworthy results?

This manuscript reports the negative correlation between taxonomic and functional diversity along a pH gradient, which is tied to bacterial genome size. Furthermore, they demonstrate environmental selection for relevant functional genes. Both of these findings are highly relevant for current and future interests in environmental microbiology, and the framework by which they conducted the study will also be highly useful for conducting this type of analysis along other environmental gradients.

2. Will the work be of significance to the field and related fields? How does it compare to the established literature? If the work is not original, please provide relevant references.

This work is significant to the field as it provided strong evidence of the suspected antithetical relationship between microbial taxonomic and functional diversity and provides a strong framework by which to evaluate the relationship. In addition, it advances our understanding of the selectivity of the individual environments studied and further investigates the consequences of the selectivity by addressing key biochemical pathways. In effect, this paper is a natural continuation of work that has been previously published, but furthers the analysis in an elegant and useful manner.

3. Does the work support the conclusions and claims, or is additional evidence needed?

The work thoroughly supports the conclusions and claims.

4. Are there any flaws in the data analysis, interpretation and conclusions? Do these prohibit publication or require revision?

I see no flaws in the analysis and appreciate the authors' attention to detail regarding the specific bioinformatic pipelines used. The thoroughness of the methods ensures reproducibility of the results, and also allows the manuscript to also serve as a reference for the specific analysis framework the authors' developed.

5. Is the methodology sound? Does the work meet the expected standards in your field?

Yes, I believe the methodology is sounds and surpasses the standards in the field. As mentioned above, I found the thoroughness of the methods section to be enlightening and appreciate the authors' effort.

6. Is there enough detail provided in the methods for the work to be reproduced?

Yes, the level of detail included in the methods of this manuscript will allow for reproduction/replication of the analysis pipeline.

Specific comments:

As detailed above, I was thoroughly impressed with this manuscript from both the results-oriented standpoint, and the usefulness of the analysis pipeline. I have no major issues with the manuscript as is, and feel the work is highly relevant and likely to advance the use of these types of analysis in the future.

However, I do suggest the authors revisit the manuscript with a Native-English Editor as there are some deficiencies in grammar and spelling that need attention prior to publication. These issues did not hinder the comprehension of the manuscript except in a few cases, but are still vital for the suitability of the manuscript for publication.

Response: Again, we appreciate the reviewer's advice about English usage and we have thoroughly edited our manuscript for English usage and clarity.

The figures for this manuscript are well-designed, but it is my suggestion to revise the color-scheme used for at least figure 5. The lime/light green color was hard to read against the blue background, even when the full-size figure was observed on a high-resolution computer monitor. Please change the color-scheme for this and figure 4 which both predominantly feature the contrast of lime-green and dark blue.

Response: We thank the reviewer for the suggestion to improve the readability of our figures. We now improved the Fig. 4 and 5 as follows.

Fig. 4

Fig. 5

Lastly, please check capitalization and punctuation in the references section as there are some inconsistencies in sentence case.

Response: Thank you for pointing out errors in references. We now have checked and thoroughly revised the capitalization and punctuation in the references section.

References in this document

- 1 Kerfahi, D. *et al.* Elevation trend in bacterial functional gene diversity decouples from taxonomic diversity. *Catena* **199**, doi:10.1016/j.catena.2020.105099 (2021).
- 2 Jordan, C. F. *Nutrient cycling in tropical forest ecosystems*. (John Wiley and Sons, 1985).
- 3 Vitousek, P. M. & Sanford, R. L. Nutrient cycling in moist tropical forest. *Ann. Rev. Ecol. Syst.* **17**, 137-167, doi:10.1146/annurev.es.17.110186.001033 (1986).
- 4 Oliverio, A. M. *et al.* The role of phosphorus limitation in shaping soil bacterial communities and their metabolic capabilities. *mBio*. **11**, e01718-01720, doi:10.1128/mBio.01718-20 (2020).
- 5 Bahram, M. *et al.* Structure and function of the global topsoil microbiome. *Nature* **560**, 233-237, doi:10.1038/s41586-018-0386-6 (2018).
- 6 Brewer, T. E., Handley, K. M., Carini, P., Gilbert, J. A. & Fierer, N. Genome reduction in an abundant and ubiquitous soil bacterium 'Candidatus Udaeobacter copiosus'. *Nat. Microbiol.* **2**, doi:10.1038/nmicrobiol.2016.198 (2017).
- 7 Malik, A. A. *et al.* Land use driven change in soil pH affects microbial carbon cycling processes. *Nat. Commun.* **9**, 3591, doi:10.1038/s41467-018-05980-1 (2018).
- 8 Ramoneda, J. *et al.* Building a genome-based understanding of bacterial pH preferences. *Sci. Adv.* **9**, eadf8998, doi:10.1126/sciadv.adf8998 (2023).

Reviewer #1 (Remarks to the Author):

First of all, I would like to thank the authors of this work for having taken a very thorough revision to the study. All my comments regarding the generalization of the results have been addressed, and they certified that the taxa along the pH gradient covered in the study were indeed adapted to those pH conditions based on the presence of genes with well-known involvement in pH adaptation. See below additional aspects of the work that need further refinement. I believe the work is now solid and interesting enough for publication. However, even though the text has improved, there are still grammatical incorrections (I showcase just a few), and a lot of the structuring of the text needs improvement to make it all more readable. This will help the study reach a larger audience and be more impactful.

Major comment on GC content: Based on the disparity in the results around GC content I recommend removing this analysis from the study. Genomic GC content reflects complex adaptations for N uptake that are not fully covered in this study, and there is no basis to think that soil pH should have an influence on GC content. Please also remove GC content from the general conceptualization on life history strategies presented in supplementary figure 19, as it is not a very good indicator of copiotrophy and generally correlates positively with genome size.

Abstract. I noted several issues in this section that should exemplify problems I could not exhaustively cover across the rest of the document and need careful attention:

L19. "enables". Gene repertoire does not enable ecosystem functioning, but one could say it "contributes to" it. Here and throughout, please revise statements that are either too vague or imprecise.

L21-22. "add". The authors of the study did not add functional gene and taxonomic diversity (add where/to what??). This is another case of imprecise writing. I would phrase this statement in a more question-driven way: "Here, we investigate mismatches between functional and taxonomic diversity..." – along those lines, catching the eye of the reader as there is a very clear question to address.

L26. "neutral; as a result,...". Example of text that reads poorly due to incorrect punctuation. Why not "...neutral. This results in bacterial taxonomic and functional diversity being negatively correlated" – along those lines. I would pass the text to a professional scientific writer to improve aspects like this one that are quite present throughout the text.

L27-28. "enriched for". The correct form is "enriched in" – this is a grammatical incorrection suggesting the authors have not taken a thorough revision to the text as pointed out in the rebuttal letter. Please take a thorough revision.

L30-32. The final sentence of the abstract does not clearly follow from the summary presented above. It is not clear to the reader why these findings alter current perceptions around microbial diversity, because the authors have not posed a question first, so the sentence is hanging there. Cohesion is another aspect that should be checked.

L111. "acid forests and neutral forests". The authors might be referring to forests with low pH and neutral pH soils.

Conclusions: I think a general conclusion that is not made explicit to the reader is that the decoupling between taxonomic and functional diversity can happen when environmental factors (such as pH) select for life history strategies that influence genome size distributions. It would be useful to include something along those lines in the discussion/conclusions. Of course, changes to genome size at the community level derive from taxonomic changes along these environmental gradients, which as the authors nicely showed respond to specific functional adaptations. Instead of focusing the conclusions on diversity across pH gradients, I think this statement is far more general in microbial ecology and well-supported finding of this study.

REVIEWERS' COMMENTS

Reviewer #1 (Remarks to the Author):

First of all, I would like to thank the authors of this work for having taken a very thorough revision to the study. All my comments regarding the generalization of the results have been addressed, and they certified that the taxa along the pH gradient covered in the study were indeed adapted to those pH conditions based on the presence of genes with well-known involvement in pH adaptation. See below additional aspects of the work that need further refinement. I believe the work is now solid and interesting enough for publication. However, even though the text has improved, there are still grammatical incorrections (I showcase just a few), and a lot of the structuring of the text needs improvement to make it all more readable. This will help the study reach a larger audience and be more impactful.

Response: We appreciate that the reviewer agrees with our revisions and gives further comments to improve our writing.

Major comment on GC content: Based on the disparity in the results around GC content I recommend removing this analysis from the study. Genomic GC content reflects complex adaptations for N uptake that are not fully covered in this study, and there is no basis to think that soil pH should have an influence on GC content. Please also remove GC content from the general conceptualization on life history strategies presented in supplementary figure 19, as it is not a very good indicator of copiotrophy and generally correlates positively with genome size.

Response: We appreciate the reviewer's suggestion on removing the disparate results of GC content. As suggested by the reviewer, to avoid overinterpretation of the results, we removed the GC content from the conceptual model in Fig. 6 (the supplementary figure 19 in the last version).

However, we prefer to keep GC content in the Results section for two reasons. First, the disparate results of GC content along pH gradient are not only detected for

our study, but also the reanalysis of Baharm et al. (2018), as presented in Supplementary Fig. 6. Second, previous understanding on the driver of GC content remains lacking, and we found that GC content was correlated with a number of biotic and abiotic variables leading by available Ca, plant richness, latitude, and total phosphorus. Together, given the importance of GC content in microbial traits, we think it worth to keep our results, despite of the disparity between methods.

Added text

Lines 129-131: Besides, both bacterial average genome size and GC% are correlated with several biotic and abiotic variables leading by available Ca (Supplementary Fig. 8).

Abstract. I noted several issues in this section that should exemplify problems I could not exhaustively cover across the rest of the document and need careful attention:

Response: The abstract was revised by the Editor, and we also checked similar problems in the rest of the text and revised accordingly.

L19. “enables”. Gene repertoire does not enable ecosystem functioning, but one could say it “contributes to” it. Here and throughout, please revise statements that are either too vague or imprecise.

Response: We have revised accordingly. Please see Lines 19-20: Bacterial gene repertoires reflect adaptive strategies, contribute to ecosystem functioning and are limited by genome size.

L21-22. “add”. The authors of the study did not add functional gene and taxonomic diversity (add where/to what??). This is another case of imprecise writing. I would phrase this statement in a more question-driven way: “Here, we investigate mismatches between functional and taxonomic diversity...” – along those lines, catching the eye of the reader as there is a very clear question to address.

Response: We have revised this sentence. Please see Lines 22-25: Here, we analyse

gene functional diversity (by shotgun metagenomics) and taxonomic diversity (by 16S rRNA gene amplicon sequencing) to investigate soil bacterial communities along a natural pH gradient in 12 tropical, subtropical, and temperate forests.

L26. “neutral; as a result,...”. Example of text that reads poorly due to incorrect punctuation. Why not “...neutral. This results in bacterial taxonomic and functional diversity being negatively correlated” – along those lines. I would pass the text to a professional scientific writer to improve aspects like this one that are quite present throughout the text.

Response: The punctuation of this sentence was revised. Please see Lines 25-28: We find that bacterial average genome size and gene functional diversity decrease, whereas taxonomic diversity increases, as soil pH rises from acid to neutral; as a result, bacterial taxonomic and functional diversity are negatively correlated.

L27-28. “enriched for”. The correct form is “enriched in” – this is a grammatical incorrection suggesting the authors have not taken a thorough revision to the text as pointed out in the rebuttal letter. Please take a thorough revision.

Response: Thanks a lot. We have substituted “enriched for” by “enriched in”. Please see Lines 28-31: The gene repertoire of acid-adapted oligotrophs is enriched in functions of signal transduction, cell motility, secretion system, and degradation of complex compounds, while that of neutral pH-adapted copiotrophs is enriched in functions of energy metabolism and membrane transport.

L30-32. The final sentence of the abstract does not clearly follow from the summary presented above. It is not clear to the reader why these findings alter current perceptions around microbial diversity, because the authors have not posed a question first, so the sentence is hanging there. Cohesion is another aspect that should be checked.

Response: We appreciate the suggestion for checking cohesion about our writing. We have revised accordingly. Please find the final sentence of the revised abstract in

Lines 31-33: Our results indicate that a mismatch between taxonomic and functional diversity can arise when environmental factors (such as pH) select for adaptive strategies that affect genome size distributions.

L111. “acid forests and neutral forests”. The authors might be referring to forests with low pH and neutral pH soils.

Response: We have revised this sentence. Please see Lines 110-113: First, principal coordinate (Pco) analysis showed that both bacterial taxonomic and functional gene compositions were divergent between forests with acid pH soils and forests neutral pH soils.

Conclusions: I think a general conclusion that is not made explicit to the reader is that the decoupling between taxonomic and functional diversity can happen when environmental factors (such as pH) select for life history strategies that influence genome size distributions. It would be useful to include something along those lines in the discussion/conclusions. Of course, changes to genome size at the community level derive from taxonomic changes along these environmental gradients, which as the authors nicely showed respond to specific functional adaptations. Instead of focusing the conclusions on diversity across pH gradients, I think this statement is far more general in microbial ecology and well-supported finding of this study.

Response: We agree with this comment. We have added relevant summary of our results in the Discussion section. Please see Lines 285-294: Our study shows that decoupling between taxonomic and functional diversity can happen when environmental factors (such as pH) select for life history strategies that influence genome size distributions. The detected changes to genome size at the community level derive from taxonomic changes along the pH gradient, i.e., a gain of small genome taxa and a loss of large genome taxa from acid to neutral pH. This taxonomic change showed responses to specific functional adaptations along a pH gradient, where bacterial taxa in acid pH soils are enriched for functions of signal transduction, cell motility, secretion system, and degradation of complex compounds, but bacterial

taxa in neutral pH soils are enriched for functions of energy metabolism and membrane transport.

Other Revisions:

Line 115: add “Supplementary Data 1”

Lines 117-118: “16S rRNA gene amplicons (R = -0.674, P < 0.001) and metagenomes (R = -0.777, P < 0.001)” <- “16S rRNA gene amplicons (R = -0.674, P = 3.925e-06) and metagenomes (R = -0.777, P = 1.52e-08)”

Lines 195-197: “Our result considers all types of genes and is in line with a recent report by Kerfahi, et al. 19, who found that microbial functional gene diversity was negatively correlated with soil pH along a gradient of increasing altitude.” <- “Our result, including all types of genes, is in line with a recent report by Kerfahi, et al. 19, who found that microbial functional gene diversity was negatively correlated with soil pH along a gradient of increasing altitude.”

Line 238 and Line 248: “Supplementary Table 3” <- “Supplementary Data 2”

Line 256: [e.g., acid] <- (e.g., acid)

Line 309: “Supplementary Fig. 19” <- “Fig. 6”

Figures legends: added description about statistical method (two-sided or one-sided); defined grey area and added “Source data are provided as a Source Data file.” if relevant.